# CCA Merge: Merging Many Neural Networks with Canonical Correlation Analysis

## Abstract

Combining the predictions of multiple trained models through ensembling is generally a good way to improve accuracy by leveraging the different learned features of the models, however it comes with high computational and storage costs. Model fusion, the act of merging multiple models into one by combining their parameters reduces these costs but doesn't work as well in practice. Indeed, neural network loss landscapes are high-dimensional and non-convex and the minima found through learning are typically separated by high loss barriers. Numerous recent works have been focused on finding permutations matching one network features to the features of a second one, lowering the loss barrier on the linear path between them in parameter space. However, permutations are restrictive since they assume a one-to-one mapping between the different models' neurons exists. We propose a new model merging algorithm, CCA Merge, which is based on Canonical Correlation Analysis and aims to maximize the correlations between linear combinations of the model features. We show that our method of aligning models leads to lower accuracy barriers when averaging model parameters than past methods. We also extend this analysis into the harder many models setting where more than 2 models are merged, and we find that CCA Merge works significantly better in this setting than past methods.

## 1 Introduction

A classical idea for improving the predictive performance and robustness of machine learning models is to use multiple trained models simultaneously. Each model might learn to extract different, complementary pieces of information from the input data, and combining the models would take advantage of such pieces; thus, benefiting the final performance (Ho, 1995). One of the simplest ways to implement this idea is to use ensembles, where multiple models are trained on a given task and their outputs are combined through averaging or majority vote at inference (Ho, 1995; Lobacheva et al., 2020). While this method yields good results, it comes with the disadvantages, particularly in the case of neural networks, of having to store in memory the parameters of multiple models and having to run each of them individually at inference time, resulting in high storage and computational costs. Another way of leveraging multiple models to improve predictive performance is to combine the different sets of parameters into a single model. This is typically done through averaging or interpolation in the parameter space of the models. After this "fusion" only a single model remains which will be used at inference time; therefore, the storage and computational costs are minimized, being the same as for a single model. The downside of such model fusion methods is that existing methods are not robust and typically do not perform as well in practice as ensembling (Stoica et al., 2023). Neural networks (NNs) are highly over-parameterized for the task they solve (Arpit et al., 2017) and their loss landscapes are high-dimensional and non-convex objects which are still somewhat poorly understood despite many recent works shedding light on some of their characteristics (Goodfellow & Vinyals, 2015; Keskar et al., 2017; Li et al., 2018; Horoi et al., 2022). Multiple good local minima can be found for a given model and task but these minima are most often separated by regions of high loss (Frankle et al., 2020). Therefore, combining the parameters from multiple trained models without falling into one of these high-loss regions and destroying the features learned during training is a hard task and constitutes an active area of research.

Previous works established empirically that any two minima in an NN parameter space found through SGD and its variants are linked by a low-loss path (Garipov et al., 2018; Draxler et al.,

2018). The term *mode connectivity* describes this phenomenon. However, to find this low-loss path between two minima one needs to run a computationally expensive optimization algorithm. As such, model fusion based on nonlinear mode connectivity has not been explored. On the other hand, *linear mode connectivity* which describes two optima connected by a *linear* low loss path in the parameter space (Frankle et al., 2020), provides a straightforward way of combining these models. Indeed, if the loss remains low on the linear path from one model to the other, merging the two models is as simple as averaging or linearly interpolating their parameters. This has emerged as a simple, yet powerful way to compare NN minima. However, this phenomenon is very rare in practice and is not guaranteed even for networks with the same initializations (Frankle et al., 2020).

One reason linear mode connectivity is hard to obtain is due to the well-known NN invariance to permutations. Indeed, it is possible to permute the neurons of an NN layer without changing the actual function learned by the model as long as the connections/weights to the subsequent layer are permuted consistently. Therefore, it is possible to have the same features learned at every layer of two different NN models and not be linearly mode-connected if the order of the features differs from one network to the other. Using this invariance as justification, Entezari et al. (2022) conjectured that most SGD solutions can be permuted in such a way that they are linearly mode connected to most other SGD solutions and presented empirical support for this conjecture. Many works in recent years have provided algorithms for finding permutations that successfully render pairs of SGD solutions linearly mode connected, or at least significantly lower the loss barrier on the linear path between these solutions, further supporting this conjecture Peña et al. (2023); Ainsworth et al. (2023); Stoica et al. (2023).

While these algorithms and the found transformations have been successful in lowering the loss barrier when interpolating between pairs of SGD solutions most of them do not consider the possibility that perhaps other linear transformations, besides permutations, would provide an even better matching of NN weights. While the permutation conjecture is enticing given it's simplicity and NN's invariance to permutations, there is nothing inherently stopping NNs from distributing computations that are done by one neuron in a model to be done by multiple neurons in another model. Permutations would fail to capture this relationship since it is not a one-to-one mapping between features. Furthermore, the focus of recent works has been mainly on merging pairs of models, and merging multiple models has received limited study. However, if a similar function is learned by networks trained on the same task then it should be possible to extract the commonly learned features from not only two but also a larger population of models. Model merging algorithms should therefore be able to find these features and the relationship between them and then merge many models without negatively affecting performance.

**Contributions** In this work we introduce CCA Merge, a more flexible way of merging models based on maximizing the correlation between linear combinations of neurons. Furthermore we focus on the difficult setting of merging not only two but also *multiple* models which were *fully trained from random initializations*. Our main contributions are threefold:

- We propose a new model merging method based on Canonical Correlation Analysis which we will refer to as "CCA Merge". This method is more flexible than past, permutation-based methods and therefore makes better use of the correlation information between neurons.

- We compare our CCA Merge to past works and find that it yields smaller accuracy barriers between interpolated models across a variety of architectures and datasets.

- We take on the difficult problem of aligning features from multiple models and then merging them. We find that CCA Merge is significantly better at finding the common learned features from populations of NNs and aligning them, leading to lesser accuracy drops as the number of models being merged increases.

## 2 RELATED WORK

**Mode connectivity** Freeman & Bruna (2017) proved theoretically that one-layered ReLU neural networks have asymptotically connected level sets, i.e. as the number of neurons increases two minima of such a network are connected by a low loss path in parameter space. Garipov et al. (2018) and Draxler et al. (2018) explore these ideas empirically and introduce the concept of *mode connectivity* to describe ANN minima that are connected by nonlinear paths in parameter space along

which the loss remains low, the maximum of the loss along this path was termed the *energy barrier*, and both works proposed algorithms for finding such paths. Garipov et al. (2018) further proposed *Fast Geometric Ensembling* (FGE) as a way to take advantage of mode connectivity by ensembling multiple model checkpoints from a single training trajectory. Frankle et al. (2020) introduced the concept of *linear mode connectivity* which describes the scenario in which two ANN minima are connected by a *linear* low loss path in parameter space. They used linear mode connectivity to study network stability to SGD noise (i.e. different data orders and augmentations). They found that at initialization, networks are typically not stable, i.e. training with different SGD noise from a random initialization typically leads to optima that are not linearly mode connected, however early in training models become stable to such noise.

**Model merging**    More recently, Entezari et al. (2022) have conjectured that "Most SGD solutions belong to a set $\mathcal{S}$ whose elements can be permuted in such a way that there is no barrier on the linear interpolation between any two permuted elements in $\mathcal{S}$" or in other words most SGD solutions are linearly mode connected provided the right permutation is applied to match the two solutions. They then perform experiments that support this conjecture and also empirically establish that increasing network width, decreasing depth, using more expressive architectures, or training on a simpler task all help linear mode connectivity by decreasing the loss barrier between two optima. Many other works seem to support this conjecture. For example, Singh & Jaggi (2020) and Peña et al. (2023) propose optimal transport-based methods for finding the best transformation to match two models. The method of Peña et al. (2023) is differentiable and they relax the constraint of finding a binary permutation method but add an entropy regularizer instead. Ainsworth et al. (2023) propose an algorithm for finding the optimal permutation for merging models based on the distances between the weights of the models themselves. Jordan et al. (2023) exposes the variance collapse phenomenon where interpolated deep networks suffer a variance collapse in their activations leading to poor performance. They propose REPAIR which mitigates this problem by rescaling the preactivations of interpolated networks through the recomputation of BatchNorm statistics.

**"Easy" settings for model averaging**    Linear mode connectivity is hard to achieve in deep learning models. Frankle et al. (2020) established that even models being trained on the same dataset with the same learning procedure and even the same initialization might not be linearly mode connected if they have different data orders/augmentations. It seems that only models that are already very close in parameter space can be directly combined through linear interpolation. This is the case for snapshots of a model taken at different points during its training trajectory (Garipov et al., 2018; Izmailov et al., 2018) or multiple fine-tuned models with the same pre-trained initialization (Wortsman et al., 2022; Ilharco et al., 2023). This latter setting is the one typically considered in NLP research. Another setting that is worth mentioning here is the "federated learning" inspired one where models are merged every couple of epochs during training (Jolicoeur-Martineau et al., 2023). The common starting point in parameter space and the small number of training iterations before merging make LMC easier to attain.

We emphasize that these settings are different from ours in which we aim to merge *fully trained models* with different parameter initializations and SGD noise (data order and augmentations).

**Merging multiple models**    Merging more than two models has only been explored thoroughly in the "easy" settings stated above. For example Wortsman et al. (2022) averages models fine-tuned with different hyperparameter configurations and finds that this improves accuracy and robustness. Jolicoeur-Martineau et al. (2023) averages the weights of a population of neural networks multiple times during training, leading to performance gains. On the other hand, works that have focused on providing feature alignment methods to be able to merge models in settings in which LMC is not trivial have only done so for 2 models at the time or for populations of simpler models such as MLPs trained on MNIST (Singh & Jaggi, 2020; Ainsworth et al., 2023; Peña et al., 2023; Jordan et al., 2023). Git Re-Basin (Ainsworth et al., 2023) proposes a "Merge Many" algorithm for merging a set of multiple models by successively aligning each model to the average of all the other models. In the appendix they use it to merge up to 32 models but only in the simple set-up of MLPs on MNIST. Singh & Jaggi (2020) also consider merging multiple models but again, only MLPs on MNIST and they go up to 4 models. We extend this line of work to more challenging settings, using more complex model architectures and larger datasets, and make this a key focus in our work.

**CCA in deep learning**   Canonical Correlation Analysis is a very popular statistical method used in many fields of sciences De Bie et al. (2005). In the context of deep learning, CCA has been used to align and compare learned representations in deep learning models Raghu et al. (2017); Morcos et al. (2018), a task which is very similar to the feature matching considered by model merging algorithms.

## 3   USING CCA TO MERGE MODELS

### 3.1   MERGING MODELS: PROBLEM DEFINITION

Let $\mathcal{M}$ denote a standard MLP and layer $L_i \in \mathcal{M}$ denote a linear layer with a $\sigma = \text{ReLU}$ activation function, weights $W_i \in \mathbb{R}^{n_i \times n_{i-1}}$ and bias $b_i \in \mathbb{R}^{n_i}$. Its input is the vector of embeddings from the previous layer $x_{i-1} \in \mathbb{R}^{n_{i-1}}$ and its output can be described as:

$$x_i = \sigma(W_i x_{i-1} + b_i)$$

Consider two deep learning models $\mathcal{A}$ and $\mathcal{B}$ with the same architecture. Let $\{L_i^{\mathcal{M}}\}_{i=1}^{N}$ be the set of layers of model $\mathcal{M} \in \{\mathcal{A}, \mathcal{B}\}$ and let $X_i^{\mathcal{M}} \in \mathbb{R}^{m \times n_i}$ denote the set of outputs of the $i$-th layer of model $\mathcal{M}$ in response to $m$ given inputs. Given the context, we assume $X_i^{\mathcal{M}}$ is centered so that each column has a mean of 0. We are interested in the problem of merging the parameters from models $\mathcal{A}$ and $\mathcal{B}$ in a layer-by-layer fashion. In practice, it is often easier to keep model $\mathcal{A}$ fixed and to find a way to transform model $\mathcal{B}$ such that the average of their weights can yield good performance. Mathematically, we are looking for invertible linear transformations $T_i \in \mathbb{R}^{n_i \times n_i}$ which can be applied at the output level of model $\mathcal{B}$ layer $i$ parameters to maximize the "fit" with model $\mathcal{A}$'s parameters and minimize the interpolation error. The output of the transformed layer $i$ of $\mathcal{B}$ is then:

$$x_i^{\mathcal{B}} = \sigma(T_i W_i^{\mathcal{B}} x_{i-1}^{\mathcal{B}} + T_i b_i^{\mathcal{B}})$$

We therefore also need to apply the inverse of $T$ at the input level of the following layer's weights to keep the flow of information consistent inside a given model:

$$x_{i+1}^{\mathcal{B}} = \sigma(W_{i+1}^{\mathcal{B}} T_i^{-1} x_i^{\mathcal{B}} + b_{i+1}^{\mathcal{B}})$$

Depending on the model's architecture, it might not be necessary to compute transformations after each single layer, for example, skip connections preserve the representation space, and the last layers of models are already aligned by the training labels. Therefore we refer to the specific layers in a network where transformations must be computed as "merging layers". After finding transformations $\{T_i\}_{i=1}$ for every merging layer in the network we can merge the two model's parameters:

$$W_i = \frac{1}{2}(W_i^{\mathcal{A}} + T_i W_i^{\mathcal{B}} T_{i-1}^{-1}) \tag{1}$$

**Merging multiple models**   In the case where multiple models are being merged there is a simple way of extending any method which aligns features between two models to the multiple model scenario. Suppose we have a set of models $\{\mathcal{M}_i\}_{i=1}^{n}$ which we want to merge. We can pick one of them, say $\mathcal{M}_j$ for $1 \leq j \leq n$, to be the *reference model*. Then we can align the features of every other model in the set to those of the reference model and average the weights. While this "all-to-one" merging approach is quite naive it seems to work well in practice.

### 3.2   BAGROUND ON CANONICAL CORRELATION ANALYSIS

Canonical Correlation Analysis (CCA) is a statistical method aiming to find relations between two sets of random variables. Suppose we have two datasets $X$ and $Y$ both of sizes $n \times d$, where $n$ is the number of instances or samples, and $d$ is the dimensionality or the number of features. Further, suppose that these datasets are centered so that each column has a mean of 0. CCA aims to find vectors $w_X$ and $w_Y$ in $\mathbb{R}^d$ such that the projections $X w_X$ and $Y w_Y$ have maximal correlation and norm 1.

Let $S_{XX} = X^\top X$, $S_{YY} = Y^\top Y$ and $S_{XY} = X^\top Y$ denote the scatter matrix of $X$, the scatter matrix of $Y$ and the cross-scatter matrix of $X$ and $Y$ respectively. In practice, it is possible to use the

respective covariance matrix which is simply $C = \frac{1}{n-1}S$ as it will yield the same solution vectors up to multiplication by a constant. The main optimization objective of CCA can be formulated as:

$$w_X, w_Y = \underset{w_X, w_Y}{\arg\max}\, w_X^\top S_{XY} w_Y$$

$$\text{s.t.} \quad \|Xw_X\|^2 = w_X^\top S_{XX} w_X = 1, \quad \|Yw_Y\|^2 = w_Y^\top S_{YY} w_Y = 1$$

This problem can be formulated as an ordinary eigenvalue problem:

$$\begin{pmatrix} \mathbf{0} & S_{XX}^{-1} S_{XY} \\ S_{YY}^{-1} S_{YX} & \mathbf{0} \end{pmatrix} \begin{pmatrix} w_X \\ w_Y \end{pmatrix} = \lambda \begin{pmatrix} w_X \\ w_Y \end{pmatrix}$$

Which can be made symmetric by introducing the vectors $v_X = S_{XX}^{1/2} w_X$ and $v_Y = S_{YY}^{1/2} w_Y$.

$$\begin{pmatrix} \mathbf{0} & S_{XX}^{-1/2} S_{XY} S_{YY}^{-1/2} \\ S_{YY}^{-1/2} S_{YX} S_{XX}^{-1/2} & \mathbf{0} \end{pmatrix} \begin{pmatrix} v_X \\ v_Y \end{pmatrix} = \lambda \begin{pmatrix} v_X \\ v_Y \end{pmatrix}$$

We can then find $v_X$ and $v_Y$ as being the left and right singular vectors of $S_{XX}^{-1/2} S_{XY} S_{YY}^{-1/2}$ respectively. The weight vectors are therefore $w_X = S_{XX}^{-1/2} v_X$ and $w_Y = S_{YY}^{-1/2} v_Y$. If we want the dimensionality of the common space to be the same as the embedding spaces we can use the whole set of left and right singular vectors of $S_{XX}^{-1/2} S_{XY} S_{YY}^{-1/2}$:

$$U, S, V^\top = \text{SVD}(S_{XX}^{-1/2} S_{XY} S_{YY}^{-1/2})$$

$$W_X = S_{XX}^{-1/2} U \quad \text{and} \quad W_Y = S_{YY}^{-1/2} V$$

For more details we direct the reader to De Bie et al. (2005) from which this section was inspired.

### 3.3 CCA Merge: Merging models with CCA

To merge models using CCA we simply apply the CCA algorithm to the two sets of activations $X_i^{\mathcal{A}}$ and $X_i^{\mathcal{B}}$ to get the corresponding weight matrices $W_i^{\mathcal{A}}$ and $W_i^{\mathcal{B}}$. Using the framework described above of matching model $\mathcal{B}$ to model $\mathcal{A}$, which is consistent with past works, we can define $T_i = W_i^{\mathcal{B}} W_i^{\mathcal{A}^{-1}}$. This transformation can be thought of as first bringing the activations of model $\mathcal{B}$ into the common, maximally correlated space between the two models by multiplying by $W_i^{\mathcal{B}}$ and then applying the inverse of $W_i^{\mathcal{A}}$ to go from the common space to the embedding space of $\mathcal{A}$. The averaging of the parameters of model $\mathcal{A}$ and transformed $\mathcal{B}$ can then be conducted following Eq. 1.

## 4 Results

### 4.1 Experimental details

We trained VGG11 models (Simonyan & Zisserman, 2015) on CIFAR10 (Krizhevsky & Hinton, 2009) and ResNet20 models on CIFAR100 . We trained models of different widths, multiplying their original width by $w \in \{1, 2, 4, 8\}$. The models were trained either using the one-hot encodings of the labels or the CLIP (Radford et al., 2021) embeddings of the class names as training objectives. This last setting is similar to the one used by Stoica et al. (2023) and we found it to yield better learned representations and performances on the task as well as less variability between random initializations.

### 4.2 CCA's flexibility allows it to better model relations between neurons

We first aim to illustrate the limits of permutation based matching and the flexibility offered by CCA Merge. Suppose we want to merge two models, $\mathcal{A}$ and $\mathcal{B}$, at a specific merging layer, and let $\{z_i^{\mathcal{M}}\}_{i=1}^n$ denote the set of neurons of model $\mathcal{M} \in \{\mathcal{A}, \mathcal{B}\}$ at that layer. Given the activations of the two sets of neurons in response to given inputs, we can compute the correlation matrix $C$ where element $C_{ij}$ is the correlation between neuron $z_i^{\mathcal{A}}$ and $z_j^{\mathcal{B}}$. For each neuron $z_i^{\mathcal{A}}$, for $1 \le i \le n$, the distribution of its correlations with all neurons from model $\mathcal{B}$ is of key interest for the problem of model merging. If, as the permutation hypothesis implies, there exists a one-to-one mapping

between $\{z_i^{\mathcal{A}}\}_{i=1}^n$ and $\{z_i^{\mathcal{B}}\}_{i=1}^n$, then we would expect to have one highly correlated neuron for each $z_i^{\mathcal{A}}$ – say $z_j^{\mathcal{B}}$ for some $1 \leq j \leq n$ – and all other correlations $C_{ik}$, $k \neq j$, close to zero. On the other hand, if there are multiple neurons from model $\mathcal{B}$ highly correlated with $z_i^{\mathcal{A}}$, this would indicate that the feature learned by $z_i^{\mathcal{A}}$ is distributed across multiple neurons in model $\mathcal{B}$ – a relationship that CCA Merge would capture.

In the top row of Fig. 1, we plot the distributions of the correlations between two ResNet20x8 models (i.e., all the elements from the correlation matrix $C$) for 3 different merging layers, one at the beginning, one in the middle and one at the end of the models. The vast majority of correlations have values around zero, as expected, since each layer learns multiple different features. In the bottom row of Fig. 1 we use box plots to show the values of the top 5 correlation values across all $\{z_i^{\mathcal{A}}\}_{i=1}^n$. For each neuron $z_i^{\mathcal{A}}$, we select its top $k$-th correlation from $C$ and we plot these values for all neurons $\{z_i^{\mathcal{A}}\}_{i=1}^n$. For example, for $k = 1$, we take the value $\max_{1 \leq j \leq n} C_{ij}$, for $k = 2$ we take the second largest value from the $i$-th row of $C$, and so on. We observe the top correlations values are all relatively high but none of them approaches full correlation (i.e., value of one), suggesting that the feature learned by each neuron $z_i^{\mathcal{A}}$ from model $\mathcal{A}$ is distributed across multiple neurons from $\mathcal{B}$ – namely, those having high correlations – as opposed to having a single highly correlated match.

Given the flexibility of CCA Merge, we expect it to better capture these relationships between the neurons of the two networks. We recall that CCA Merge computes a linear transformation $T = W_{\mathcal{B}} W_{\mathcal{A}}^{-1}$ that matches to each neuron $z_i^{\mathcal{A}}$ a linear combination $z_i^{\mathcal{A}} \approx \sum_{j=1}^n T_{ij} z_j^{\mathcal{B}}$ of the neurons in $\mathcal{B}$. We expect the distribution of the coefficients (i.e., elements of $T$) to match the distribution of the correlations ($C_{ij}$ elements), indicating the linear transformation found by CCA Merge adequately models the correlations and relationships between the neurons of the two models. For each neuron $z_i^{\mathcal{A}}$, we select its top $k$-th, for $k \in \{1, 2\}$, correlation from the $i$-th row of $C$ and its top $k$-th coefficient from the $i$-th row of $T$ and we plot a histogram of these values for all neurons $\{z_k^{\mathcal{A}}\}_{k=1}^n$ in Fig. 2. Indeed, the distributions of the correlations and those of the CCA Merge coefficients are visually similar, albeit not fully coinciding. To quantify this similarity we compute the Wasserstein distance between these distributions, normalized by the equivalent quantity if the transformation were a permutation matrix. For a permutation matrix, the top 1 values would be of 1 for every neuron $z_i^{\mathcal{A}}$ and all other values would be 0. We can see that CCA Merge finds coefficients that closely match the distribution of the correlations, more so than simple permutations, since the ratio of the two distances are 0.15, 0.04, and 0.08, respectively, for top 1 values in the three considered layers. Top 2 relative distances are a bit higher at 0.35, 0.23 and 0.28 respectively since the correlations distribution is closer to zero than to one therefore closer to top 2 permutation coefficients, however it is still the case that the CCA Merge coefficients have a distribution more closely matching that of the correlations.

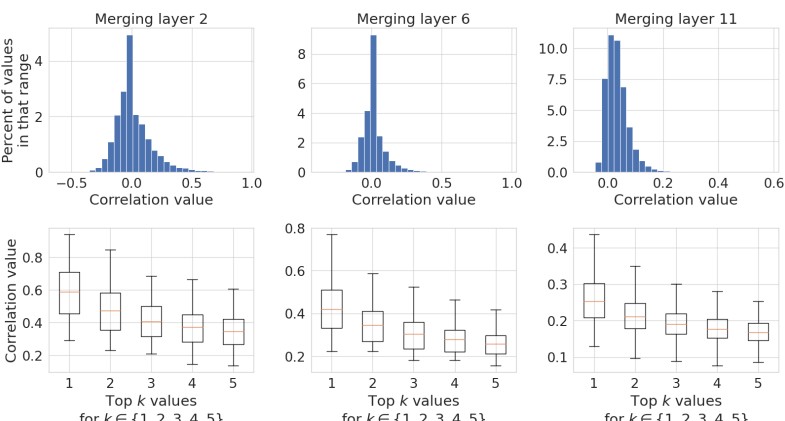

Figure 1: **Top row:** distribution of correlation values between the neurons $\{z_i^{\mathcal{A}}\}_{i=1}^n$ and $\{z_i^{\mathcal{B}}\}_{i=1}^n$ of two ResNet20x8 models ($\mathcal{A}$ and $\mathcal{B}$) trained on CIFAR100 at three different merging layers; **Bottom row:** for $k \in \{1, 2, 3, 4, 5\}$ the distributions of the top $k$-th correlation values for all neurons in model $\mathcal{A}$ at those same merging layers.

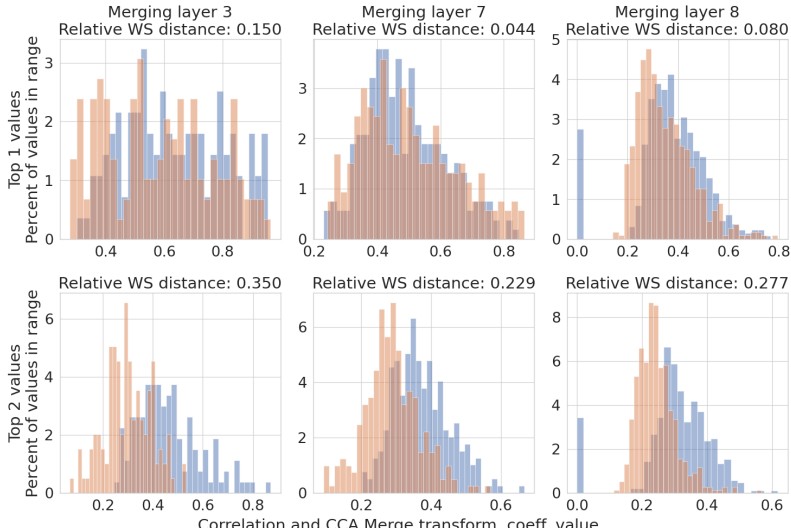

Figure 2: Distributions of top 1 (**top row**) and 2 (**bottom row**) correlations (blue) and CCA Merge transformation coefficients (orange) across neurons from model $\mathcal{A}$ at three different merging layers. In the top row for example, for each neuron $z_i^{\mathcal{A}}$ we have one correlation value corresponding to $\max_{1 \leq j \leq n} C_{ij}$ and one coefficient value corresponding to $\max_{1 \leq j \leq n} T_{ij}$ where $C$ is the cross-correlation matrix between neurons of models $\mathcal{A}$ and $\mathcal{B}$, and $T$ is the CCA Merge transformation matching neurons of $\mathcal{B}$ to those of $\mathcal{A}$. Wasserstein distance between the distributions of top $k \in \{1, 2\}$ correlations and top $k$ Merge CCA coefficients are reported, relative to equivalent distances between correlations and Permute transforms (all top 1 values are 1, and top 2 values are 0).

### 4.3 MODELS MERGED WITH CCA MERGE PERFORM BETTER

In Table 1 we show the test accuracies of merged VGG11 models of different widths trained on CIFAR10 for CCA Merge and multiple other popular model merging methods. The number of models being merged is 2 and for each experiment, we report the mean and standard deviation across 4 merges where the base models were trained with different initialization, data augmentation, and data order seeds. We report the average accuracies of the base models being merged ("Base models avg.") as well as the accuracies of ensembling the models, which is considered to be the upper limit of what model fusion methods can achieve. We compare CCA Merge with the following model merging methods:

- **Direct averaging:** directly averaging the models' weights without applying any transformation to align neurons.
- **Permute:** permuting model weights to align them, the permutation matrix is found by solving the linear sum assignment problem consisting of maximizing the sum of correlation between matched neurons. This method is equivalent to the "Matching Activations" method from Ainsworth et al. (2023), the "Permute" method considered in Stoica et al. (2023) and the neuron alignment algorithm proposed by Li et al. (2015) and used in Jordan et al. (2023).
- **Matching Weights:** permuting model weights by directly minimizing the distance between the two model weights. This is the main method from Ainsworth et al. (2023).
- **ZipIt!:** model merging method proposed by Stoica et al. (2023). We note that comparing to this method is somewhat of an unfair comparison since ZipIt! also allows the merging of neurons from the same network which results in a redundancy reduction effect that the other methods do not have. Also, it isn't strictly speaking a permutation-based method although similar.

In all cases, we apply REPAIR (Jordan et al., 2023) to recompute the BatchNorm statistics after the weight averaging and before evaluating the merged model to avoid variance collapse.

VGG11 models merged with CCA Merge have significantly higher accuracies than models merged with any other method, and this is true across all model widths considered. Differences in accuracy ranging from 10% ($\times 8$ width) up to 25% ($\times 1$ width) can be observed between CCA Merge and the

Table 1: VGG11/CIFAR10 - Accuracies and standard deviations after merging 2 models

| Width multiplier | $\times 1$ | $\times 2$ | $\times 4$ | $\times 8$ |
|---|---|---|---|---|
| Base models avg. | $87.27 \pm 0.25\%$ | $87.42 \pm 0.86\%$ | $87.84 \pm 0.21\%$ | $88.20 \pm 0.45\%$ |
| Ensemble | $89.65 \pm 0.13\%$ | $89.74 \pm 0.44\%$ | $90.12 \pm 0.16\%$ | $90.21 \pm 0.24\%$ |
| Direct averaging | $10.54 \pm 0.93\%$ | $10.28 \pm 0.48\%$ | $10.00 \pm 0.01\%$ | $10.45 \pm 0.74\%$ |
| Permute | $54.74 \pm 6.41\%$ | $63.01 \pm 1.18\%$ | $64.67 \pm 1.63\%$ | $62.58 \pm 3.07\%$ |
| ZipIt! | $53.15 \pm 8.08\%$ | $60.73 \pm 2.07\%$ | - | - |
| Matching Weights | $55.40 \pm 4.67\%$ | $66.98 \pm 1.96\%$ | $71.92 \pm 2.21\%$ | $73.74 \pm 1.77\%$ |
| CCA Merge | $81.01 \pm 1.98\%$ | $83.36 \pm 0.96\%$ | $84.94 \pm 0.61\%$ | $84.30 \pm 2.27\%$ |

Table 2: ResNet20/CIFAR100 - Accuracies and standard deviations after merging 2 models

| Width multiplier | $\times 1$ | $\times 2$ | $\times 4$ | $\times 8$ |
|---|---|---|---|---|
| Base models avg. | $69.17 \pm 0.25\%$ | $74.17 \pm 0.14\%$ | $77.16 \pm 0.34\%$ | $78.76 \pm 0.27\%$ |
| Ensemble | $73.47 \pm 0.23\%$ | $77.57 \pm 0.22\%$ | $79.86 \pm 0.07\%$ | $80.98 \pm 0.23\%$ |
| Direct averaging | $1.63 \pm 0.15\%$ | $2.65 \pm 0.16\%$ | $5.10 \pm 0.54\%$ | $13.94 \pm 1.65\%$ |
| Permute | $28.66 \pm 2.22\%$ | $48.96 \pm 0.36\%$ | $64.42 \pm 0.58\%$ | $72.89 \pm 0.12\%$ |
| ZipIt! | $25.36 \pm 3.01\%$ | $47.59 \pm 0.43\%$ | $63.32 \pm 0.53\%$ | $71.77 \pm 0.24\%$ |
| CCA Merge | $31.47 \pm 1.57\%$ | $54.03 \pm 0.36\%$ | $68.43 \pm 0.44\%$ | $74.76 \pm 0.05\%$ |

second-best performing method. Furthermore, CCA Merge is more robust when merging smaller width models, incurring smaller accuracy drops than other methods when the width is decreased; 3.29% drop for CCA Merge versus 18.34% for Matching Weights and 7.84% for Permute. Lastly, CCA Merge seems to be more stable across different initializations, the accuracies having smaller standard deviations than all other methods for the same width except for Matching Weights for $\times 8$ width models. We note that for model width multipliers above $\times 2$, we ran into out-of-memory issues when running ZipIt!, which is why those results are not present.

Table 2 contains similar results but for ResNet20 trained on CIFAR100. Here we only compare with ZipIt! and Permute since Jordan et al. (2023) and Stoica et al. (2023) have suggested that Permute is better than Matching Weights in this setting. Again, across all model widths, we observe that CCA Merge yields merged models having better accuracies than other methods, although the differences here are less pronounced. Furthermore, our method seems again to be more stable across different random seeds, having lower standard deviations than other methods. In Sec. A.1 we present results with ResNet20 models trained on disjoint subsets of CIFAR100, where CCA Merge also achieves smaller drops in accuracy than the other methods.

For both VGG and ResNet architectures as well as both considered datasets the added flexibility of CCA Merge over permutation-based methods seems to benefit the merged models. Aligning neurons from two different models using linear combinations allows CCA Merge to better model relationships between neurons and to take into account features that are distributed across multiple neurons. In addition to the raw performance benefits, CCA Merge seems to be more stable across different model widths as well as across different random initializations and data order and augmentation.

## 4.4 CCA MERGING FINDS BETTER COMMON REPRESENTATIONS BETWEEN MANY MODELS

In this section, we present our results related to the merging of many models. This constitutes the natural progression to the problem of merging pairs of models and is a significantly harder task. Furthermore, aligning populations of neural networks brings us one step closer to finding the common learned features that allow different neural networks to perform equally as well on complex tasks despite having different initializations, data orders, and data augmentations.

As previously mentioned, the problem of merging many models is often ignored by past works except for the settings in which linear mode connectivity is easily obtained. The authors of Ainsworth et al. (2023) introduced "Merge Many", an adaptation of Matching Weights for merging a set of models. In Merge Many each model in the set is sequentially matched to the average of all the others until convergence. A simpler way of extending any model merging method to the many models setting is to choose one of the models in the group as the *reference model* and to align every other

network in the group to it. Then the reference model and all the other aligned models can be merged. It is by using this "all-to-one" merging that we extend CCA Merge, Permute, and Matching Weights to the many model settings. ZipIt! is naturally able to merge multiple models since it aggregates all neurons and merges them until the desired size is obtained.

In Fig. 3 we show the accuracies of the merged models as the number of models being merged increases. For both architectures and datasets aligning model weights with CCA continues to yield lower accuracy barriers. In fact, models merged with CCA Merge applied in an all-to-one fashion maintain their accuracy relatively well while the ones merged with other methods see their accuracies drop significantly. In the VGG case, the drop in accuracy for other methods is drastic, all merged models have less than 20% accuracy when more than 3 models are being merged which constitutes a decrease of more than 25% from the 2 models merged scenario. CCA Merge on the other hand suffers a drop in accuracy of less than 3% when going from merging 2 models to 5. We also note that despite being designed specifically for the many models setting, Merge Many performs only slightly better than its 2 model counterpart Matching Weights applied in an all-to-one fashion.

For ResNets, the accuracy of models merged with Permute drops by ~15% when going from merging 2 models to 20. While less drastic than in the VGG case this decrease in performance is still significant. ZipIt! displays a slightly more pronounced drop when going from 2 models merged to 5. CCA Merge on the other hand is a lot more robust, incurring a less than 4% drop in accuracy even as the number of merged models is increased to 20. Additionally, accuracy values for models merged by CCA Merge seem to plateau sooner than those for Permute.

These results suggest that CCA Merge is significantly better than past methods at finding the "common features" learned by groups of neural networks and aligning them. The limitations of permutation-based methods in taking into account complex relationships between neurons from different models are highlighted in this context. Here it is harder to align features given that there are more of them to consider and therefore it is easier to destroy these features when averaging them.

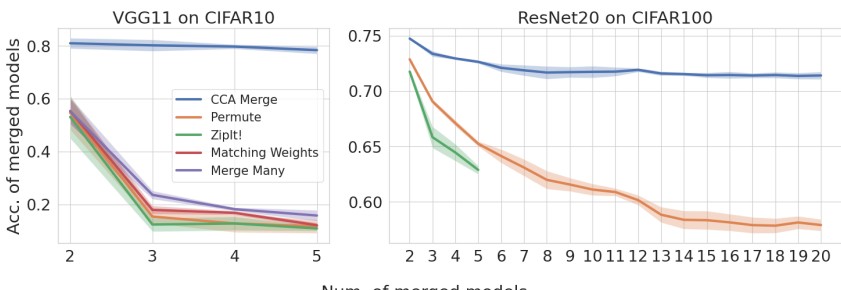

Figure 3: Accuracies of averaged models after feature alignment with different model fusion methods. The mean and standard deviation across 4 random seeds are shown.

## 5 DISCUSSION AND CONCLUSION

Recent model fusion successes exploit inter-model relationships between neurons by modelling them as permutations before combining them. Here, we argue that, while assuming a one-to-one correspondence between neurons yields interesting merging methods, it is rather limited as not all neurons from one network have an exact match with a neuron from another network. Our proposed *CCA Merge* takes the approach of linearly transforming model parameters beyond permutation-based optimization. This added flexibility allows our method to outperform recent competitive baselines when merging pairs of models (Tables 1 and 2). Furthermore, when considering the harder task of merging many models, CCA Merge models showcase remarkable accuracy stability as the number of models merged increases, while past methods suffer debilitating accuracy drops.

Merging many models successfully, without incurring an accuracy drop, is one of the big challenges in this area of research. Our method, CCA Merge, makes a step in the direction of overcoming this challenge. As future work, it would be interesting to further study the common representations learned by populations of neural networks. Further, an interesting future research avenue is to test CCA Merge in the context where the models are not trained on the same dataset, but rather on different ones, to see if it is capable of merging different learned features.

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

# A  APPENDIX

## A.1  RESULTS ON DISJOINT DATASETS

Here we present results from ResNet20 models trained on disjoint splits of the CIFAR100 dataset. We use samples from a Dirichlet distribution with parameter vector $\alpha = (0.5, 0.5)$ to subsamble each class in the dataset and create 2 disjoint data splits, one for each model to be trained on. In Table 3 we report mean and standard deviation of accuracies across 4 different model pairs, each having different random seeds for the dataset subsampling and model initializations.

Table 3: ResNet20/CIFAR100 Disjoint - Accuracies and standard deviations after merging 2 models

| Width multiplier | ×1 | ×2 | ×4 | ×8 |
|---|---|---|---|---|
| Base models avg. | $52.61 \pm 1.84\%$ | $56.78 \pm 1.96 \%$ | $58.74 \pm 1.93\%$ | $59.98 \pm 1.80\%$ |
| Ensemble | $64.68 \pm 0.76\%$ | $69.92 \pm 0.75\%$ | $72.18 \pm 0.67\%$ | $73.77 \pm 0.44\%$ |
| Direct averaging | $1.48 \pm 0.17\%$ | $3.12 \pm 0.49\%$ | $7.03 \pm 0.49\%$ | $20.55 \pm 3.07\%$ |
| Permute | $20.66 \pm 0.87\%$ | $39.05 \pm 0.42\%$ | $51.55 \pm 0.86\%$ | $58.45 \pm 1.76\%$ |
| ZipIt! | $19.24 \pm 2.00\%$ | $37.43 \pm 0.65\%$ | $50.72 \pm 0.89\%$ | $57.97 \pm 1.29\%$ |
| CCA Merge | $24.46 \pm 3.20\%$ | $44.59 \pm 1.80\%$ | $55.61 \pm 0.63\%$ | $60.38 \pm 1.68\%$ |

