# OpenReview forum: "CCA Merge: Merging Many Neural Networks with Canonical Correlation Analysis"
_ICLR.cc/2024/Conference — ICLR 2024 Conference Withdrawn Submission_

### Official Review · Reviewer_E1Xu · 2023-10-25

**Soundness:** 1 poor
**Presentation:** 2 fair
**Contribution:** 3 good
**Rating:** 6
**Confidence:** 3

**Summary:**

The authors identify a research gap in existing methods of merging different trained neural network models based on neuron permutations, and propose to employ Canonical Correlation Analysis instead. They propose a simple algorithm to align the outputs of each pair of layers, by applying the inverse CCA transforms to the layer’s weights. They conduct an exploratory analysis of the distributions of correlation coefficients of different layers’ neurons, and perform experiments on two Cifar datases, where the method outperforms the state of the art approaches.

**Strengths:**

- The method is relatively easy to implement
- The results are better than the state of the art
- The method can be applied in settings with more than 2 models
- The introduction part of the paper is written well, it was easy to understand the motivation and the research gaps

**Weaknesses:**

- A sound theoretical justification is missing - the authors provide only an intuition as to why the method would be better than the permutation-based approach
- Section 4.2 is not fully convincing as a justification as to why the approach would be sound

**Questions:**

- The main problem see is that the authors apply CCA after the nonlinearities, but apply the projection on the weight matrices, which precede the nonlinearities. Wouldn’t it make more sense to compute CCA just after the linear transformation, where we are still in the linear regime? Doesn’t this break the “flow of information” in the neural network?
- I do not fully understand the message behind Figure 2 - why would we actually want to match the distribution of correlations? Isn’t it also possible that a neuron in Neural Network B is a linear transformation of neurons in Network A? That wouldn’t be captured by a mere one-to-one correlation coefficient.
- It is unclear what “base models avg.” vs. “Ensemble” actually are. I assume the first is averaging the predictions and the second is majority voting, but please be specific in the text

---

### Official Review · Reviewer_WLcu · 2023-10-30

**Soundness:** 2 fair
**Presentation:** 2 fair
**Contribution:** 2 fair
**Rating:** 3
**Confidence:** 3

**Summary:**

This paper studied the problem of merging the parameter values (weights) of multiple neural networks. The assumption is that with two models with the same architecture, linearly combining the two sets of weights can result in a model of better prediction on unseen data. This works studies a particular setting, where there is a reference model A and the aim is to transform weights from another model B to best align with the weights of model A. The authors proposed a transformation based on Canonical Correlation Analysis (CCA). Assuming two sets of data X_A and X_B, CCA can generate a pair of transformations T_A and T_B such that T_A(X_A) and T_B(X_B) are a pair of orthonormal representations with aligned features (Note that they are not necessarily identical). Then the authors use (T_A)^{-1}T_B as the transformation to maps weights in model B into the weight space for model A. They stated that this transformation worked better than a number of baselines based on mostly simple averaging.

**Strengths:**

The related work section contains a good review of rather diverse topics.

**Weaknesses:**

The motivation is not very clear.

First of all, why do we need linear combination (or interpolation) of model weights (except in the FL setting explicitly excluded by the authors)? In the experiment section, the author should have compared with the simple baseline of directly using the best performing model.

Next, CCA does not provide a common set of coordinates. It only gives you a pair of representations. Why don't we directly learn a mapping from B to A? What theoretical and empirical justifications for going through two CCA transformations?

I believe that the meat of the method is in Section 3.3. It does not contain sufficient description of the method. The definition of T_i is incorrect, I believe, if W_i^B and W_i^A is defined as in Section 3.2

**Questions:**

- Reference [Jolicoeur 2023], where can I find that paper?

- In Section 3.3, what is the inverse of a weight matrix? How do you compose it with a weight matrix? Which transform composition convention do you adopt (left first or right first)? I see two conventions in Section 3.2 and 3.3.

---

### Official Review · Reviewer_vbxJ · 2023-10-30

**Soundness:** 3 good
**Presentation:** 2 fair
**Contribution:** 3 good
**Rating:** 6
**Confidence:** 3

**Summary:**

This paper shows the limitation of one-to-one matching of neurons and demonstrates that matching using correlation information is effective for merging models. In particular, compared to the baseline, the loss barrier can be kept small when many networks are merged.

**Strengths:**

1. The author confirms the effectiveness of the proposed method between two or more merging models.
2. The motivation for model merging by CCA beyond permutation is experimentally confirmed. Experiments confirm that one-to-one matching like permutation is not sufficient in terms of correlation.

**Weaknesses:**

1. According to the introduction, model merge aims at robust inference like ensemble without increasing inference cost. On the other hand, the motivation for merging many models is unclear, since even the proposed method is less accurate when merging many models. It is easy to see the motivation if the accuracy of ensemble as oracle is also plotted in Fig. 3.
2. There is no theoretical support for the rationale that merging models to increase correlation through CCA will lower the loss barrier more.

**Questions:**

1. Why is correlation important as a criterion for alignment? Is it not possible that the linear transformation has more degrees of freedom than permutation, and therefore matching is just working well? For example, is the proposed method more effective than weight matching using linear transformations?
2. How is the computational cost of the proposed method compared to weight matching and activation matching? I am particularly concerned about the higher cost of finding the inverse of T when the network is wide. I think speed is important given the motivation to merge many models.
3. CCA does not seem to learn the correlation 0 at layer 8 in Figure 2. Why is it so difficult for CCA to capture correlation 0?
4. Since matching weight is a strong baseline in Table 2, I would like to see matching weight results added to Table 3 as well.

---

### Official Review · Reviewer_XYpB · 2023-11-08

**Soundness:** 1 poor
**Presentation:** 2 fair
**Contribution:** 1 poor
**Rating:** 3
**Confidence:** 5

**Summary:**

This work proposes an improved algorithm variant to fuse networks trained from different random initializations by relying on CCA. This allows a more flexible approach of merging the corresponding neurons, as opposed to finding one-to-one matching or permutation. The results are demonstrated on fusing networks of increasing width as well as for multiple models.

**Strengths:**

- The proposed CCA variant seems to show promise in the presented results over the considered baselines.
- Interesting analysis to see how distributed are the neuron representations.

**Weaknesses:**

- **Incorrect claims about "CCA merge is more flexible":** The discussion around this is far from correct. Already, in one of the early works in this area (Singh & Jaggi, 2020), there is already a facility to do a more general matching via the transport map, precisely for the case when the neuron features are more distributed. In their experiments, when the networks have the same number of neurons in the layers, Optimal transport finds a solution at the extreme point (which corresponds to a permutation); however, if the number of neurons is different, the obtained Transport map will not be one-hot in each of the rows but will have a more distributed spread. Even more than that, when the number of neurons are the same, one can use their entropy-regularized version (and besides this is also differentiable, in addition to Pena et al 2022) which will also produce a more general linear transformation.

&nbsp;

- **Fusing multiple models on complex architectures and datasets has already been done:** This is yet another wrong claim. Singh & Jaggi 2020 indeed present results of fusing more number of VGG11 networks than considered in this paper, and that too on CIFAR100 instead of CIFAR10 considered here. Hence, this whole aspect is misrepresented throughout the discussion here. Likewise, the approach of choosing a reference model is, plain and simple, already discussed explicitly in Singh & Jaggi 2020. See their algorithm 1 and the several mentions of the initial estimate of the fused model.

&nbsp;

- **Questionable utility of proposed merging algorithm with poor results relative to parent networks:** In none of the experiments, does the proposed algorithm outperform the parent networks, which beats the point of combing different networks. A common approach in the literature is to further fine-tune the obtained networks post fusion. It would be important to see a comparison of the proposed approach in such a setting, compared with previous baselines as in Table 2 of Singh & Jaggi 2020. Right now, it is not clear if the networks after fusion, can even be fine-tuned to deliver a gain over the parent networks, and if CCA Merge does better than past methods like OTFusion/Git Re-Basin on this in such a setting.

**Questions:**

Check the weaknesses section.